# The Upper Limb Orthosis in the Rehabilitation of Stroke Patients: The Role of 3D Printing

**DOI:** 10.3390/bioengineering10111256

**Published:** 2023-10-27

**Authors:** Andrea Demeco, Ruben Foresti, Antonio Frizziero, Nicola Daracchi, Francesco Renzi, Margherita Rovellini, Antonello Salerno, Chiara Martini, Laura Pelizzari, Cosimo Costantino

**Affiliations:** 1Department of Medicine and Surgery, University of Parma, 43126 Parma, Italy; antonio.frizziero@unipr.it (A.F.); nicola.daracchi@unipr.it (N.D.); francesco.renzi@unipr.it (F.R.); margherita.rovellini@unipr.it (M.R.); antonello.salerno@unipr.it (A.S.); chiara.martini@unipr.it (C.M.); cosimo.costantino@unipr.it (C.C.); 2Center of Excellence for Toxicological Research (CERT), University of Parma, 43126 Parma, Italy; 3Italian National Research Council, Institute of Materials for Electronics and Magnetism (CNR-IMEM), 43124 Parma, Italy; 4AUSL Piacenza, Neurorehabilitation and Spinal Unit, Department of Rehabilitative Medicine, 29121 Piacenza, Italy; l.pelizzari@ausl.pc.it

**Keywords:** stroke, 3D printing, disability, neurologic disease, Healthcare 4.0

## Abstract

Stroke represents the third cause of long-term disability in the world. About 80% of stroke patients have an impairment of bio-motor functions and over half fail to regain arm functionality, resulting in motor movement control disorder with serious loss in terms of social independence. Therefore, rehabilitation plays a key role in the reduction of patient disabilities, and 3D printing (3DP) has showed interesting improvements in related fields, thanks to the possibility to produce customized, eco-sustainable and cost-effective orthoses. This study investigated the clinical use of 3DP orthosis in rehabilitation compared to the traditional ones, focusing on the correlation between 3DP technology, therapy and outcomes. We screened 138 articles from PubMed, Scopus and Web of Science, selecting the 10 articles fulfilling the inclusion criteria, which were subsequently examined for the systematic review. The results showed that 3DP provides substantial advantages in terms of upper limb orthosis designed on the patient’s needs. Moreover, seven research activities used biodegradable/recyclable materials, underlining the great potential of validated 3DP solutions in a clinical rehabilitation setting. The aim of this study was to highlight how 3DP could overcome the limitations of standard medical devices in order to support clinicians, bioengineers and innovation managers during the implementation of Healthcare 4.0.

## 1. Introduction

Stroke is a cerebrovascular accident that caused 5.5 million deaths in 2016 and represents one of the most important causes of disabilities in the world [1,2,3]. Moreover, considering population ageing and the rise of other modifiable risk factors, the incidence is expected to increase [3].

The sites of brain damage, as well as their extent, influence the stroke’s outcome [1]. In detail, in about 80% of cases there is an impairment of motor function linked to an altered muscle tone with a predominantly spasticity flexor. This can affect the lower limbs, trunk and upper limbs, with implications for the ability to walk and grasp objects [4].

The clinical signs could also include dysphagia, impairment in speech, behavioral functions, sensory functions, visual functions and cognitive functions such as attentional deficits, memory, orientation and awareness [1,5,6], reducing patients’ and caregivers’ quality of life [1,5,7]. In this context, rehabilitation plays a key role in the treatment of patient disabilities. In fact, it is recommended to perform a multidisciplinary evaluation (including, e.g., an occupational therapist, physical therapist and speech therapist) to address the patient needs, as well as the psychological and social context [8]. In detail, even if functional recoveries are still described six months post stroke [5], the first three months are the most important, so it is significant to set up an early, personalized and intensive rehabilitation plan [9]. In the early stages, improvement is mainly due to the autonomous recovery ability of the brain, whereas in later phases it is based on brain plasticity and cortical reorganization [1,5].

However, in chronic stroke patients, residual disability may include mobility disorder, poor balance, gait impairment and limited physical activity [10]. Consequently, to maintain and maximize functional recovery, patients with chronic stroke require long-lasting rehabilitation or exercise intervention [10].

Nevertheless, over half of stroke patients with upper limb impairments fail to regain arm functionality, resulting in a serious loss in terms of independence and social activities. Therefore, the use of the upper limbs is essential to interact with the environment and with people, and 87% of stroke patients showed hand paralysis with impairment of the clamp and grab movements and a negative impact on the activities of daily living (ADLs) [11].

In these patients, the use of upper limb orthotics or splints, devices externally applied that modifiy the structural and functional features of the muscle-skeletal system, is widely accepted [12].

Orthoses can be classified as standard or custom-made. Standard orthoses are less expensive but might provide less comfort to a patient compared to a custom-made device, as they come in limited sizes and are challenging to adapt. Alternatively, a traditional custom-made orthosis, namely, high-temperature and low-temperature thermoplastic orthosis, has the potential to enhance comfort and be more adequate, although the manufacturing process is more challenging [13]. The conventional method for manufacturing a custom orthosis is to make a mold of the patient’s anatomy, most commonly using a plaster cast. The orthosis is shaped on the mold, commonly using high-temperature thermoplastics [14]. Then, it must be manually adapted to be ready for fitting [15]. The entire process is time-consuming and may take from two days to several weeks to complete a patient’s prescription. The manufacture of custom orthoses from molds generates a large amount of waste [16].

When clinical requirements demand a simpler design, custom orthoses may be handcrafted from low-temperature thermoplastic sheets molded directly on the patient’s body. However, such orthoses come with several functional and aesthetic limitations. Patients often experience problems with proper skin ventilation and maintaining cleanliness and dryness of their splints [17].

Furthermore, conventional orthoses are not easily adaptable to morphological changes (i.e., swelling, shrinkage, changes in muscle stiffness or range of motion, pediatric patients), and modifications require highly skilled personnel and additional working hours [18,19]. Despite the benefits of traditional custom-made orthoses, complications and discomfort have been reported. These include skin lesions, poor fitting, sweating due to lack of breathability, heavy weight, inability to get wet or be cleaned, bulkiness and poor waterproofing, all of which can have a significant impact on compliance and the effectiveness of the rehabilitation treatment [20].

Moreover, since they are handmade, the risk of complications and discomfort, particularly skin breakdown and poor fit, depends largely on the skill and experience of the technician [21].

Orthoses also include exoskeletons, devices made to produce a task-oriented motor training, to assist the patient in the execution of a particular movement. The exoskeleton has a structure associated with an engine, and it works with an input; for example, the EMG surface signals from the patient’s muscles [22,23]. The use of these orthoses, combined with conventional rehabilitation, can be useful to improve the intensity of training and promote repetitive movements, thus improving the recovery of the patient [11].

In the last 20 years, new techniques have been employed in rehabilitation with the aim of personalizing clinical devices/therapies, starting from the patient’s morphology to ensure the greatest comfort during task-oriented motor training; among these, 3D printing (3DP) seems to play a key role. Three-dimensional printing or additive manufacturing consists of the construction of a three-dimensional object from a digital 3D model or a computer-made model (CAD). The production of a 3D object can be achieved through different technologies belonging to seven families: powder bed fusion, binder jetting, directed energy deposition, VAT polymerization, material jetting, material extrusion and sheet lamination.

Three-dimensional printing (3DP) transforms 3D designs into life-changing products, including patient-friendly pharmaceutical products as well as bio-inspired medical devices, resulting in the next revolution for the pharmaceutical and medical device industries (from surgery to polypharmaco-therapy, i.e., training models, prosthesis, implantable parts, human organs or tablets) [24].

Healthcare utilizes all of the AM families (ASTM F42—Additive Manufacturing) and the related 3D printing technologies, such as VAT polymerization via stereolithography (SLA) [25] or digital light processing (DLP) technologies [26,27,28]; material jetting (MJ) by using Polyjet (PJ) technology; material extrusion—adopting fused deposition modeling (FDM) [29,30] or robocasting (RC)/Direct Ink Writing (DIW); powder bed fusion (PBF) with selective laser sintering (SLS) [31] or selective laser beam melting [32,33]; binder jetting (BJ) [34] typically multi-jet printing (MJP) technology; directed energy deposition (DED), powder or wired [35] and sheet lamination (SL) [36] that, depending by the technology should became laminated object manufacturing (LOM) or selective deposition lamination (SDL).

VAT polymerization is the family commonly used to produce dental and orthopedic implants, with medical industry applications [37,38,39,40,41,42,43,44]; resins react to a specific laser wavelength, building the designed object layer-by-layer. Material extrusion 3D printers are the most diffused, user-friendly and low-cost. The fused filament (typically PLA = 20–40 €/kg, quality dependent) is heated to create the final 3D-printed object [45], or a syringe extrudes hydrogels or fluids. Both of these technologies have a wide range of materials, but with material extrusion technologies we can print more than one material in the same printing process, supporting sustainability when adopted with biodegradable materials (one piece for a patient requires a mandatory recycling approach) [46] or bioresorbable materials [47]. While VAT polymerization assures a reduced perception of the layers and validates biomedical resins and procedures, it is provided directly by the producers. Therefore, it is possible to allow doctors’ surgery planning and understand the patient’s specific anatomy [29,48] and introduce the 3D model inside the surgery room.

To overcome the aforementioned limitation, it is possible to use material jetting technologies, which offer the potential to print multi-material and color inks deposited with a layer height of about 14µm, with the related high aesthetic impact.

PBF, DED and BJ technologies selectively fuse [39,40,49] or bind [50,51] powdered materials and are often used to produce orthopedics, prosthetics or pharmaceutical applications [52,53], respectively. PBF is characterized by a fast production time and the absence of binding agents; DED fuses only the required material (binder not required), while BJ can modulate mechanical behavior and color but uses the same powder contained in the 3D printer binder.

Finally, SL, by using foils, supports the creation of 3D colored objects useful for teaching activities or phantoms [35,54].

Furthermore, 3DP applications and properties are strictly related to 3D printing technologies and natural or artificial materials (biocompatible or not) [55,56]. ABS (acrylonitrile butadiene styrene) and PLA (polylactic acid) are the most popular thermoplastic materials. They are relatively easy to print and are used to create objects such as disposable dishes, engineered prototypes [57,58] or medical devices [59,60]. Dental implants should be printed by using ceramic materials [61,62] or metals (ferrous and non-ferrous alloys, stainless steel, aluminum, titanium, cobalt–chromium alloys, gold and platinum) already used to produce joint replacements and pacemakers [63,64]. Moreover, by combining two or more materials (fiber and matrix) we obtain composite materials, enabling the customization of specific properties, such as high strength or stiffness (i.e., carbon fiber-reinforced plastics and glass fiber-reinforced plastics) [65,66].

Finally, to produce living tissue and organ bioprinting requires the combination of biological/synthetic materials [67,68,69], enabling the bioplotting of organoids by adding cells into the bio-ink [70,71].

The mechanochemical behavior of 3D-printed biomedical materials plays a pivotal role in assuring adherence to the final application functionality. These materials are exposed to mechanical stresses or chemical reactions that may influence their properties (i.e., scaffolding, phenomena detection, cellular interactions), requiring the accurate study of mechanochemical systems [72,73] and biocompatibility (cytotoxicity, in-vitro/in-vivo). The final aim has to confirm the safety, compatibility and mechanical limitations [74,75] before scaling the devices for human use and refining the design and production procedures to improve performance [74,76,77,78,79].

Therefore, to give a base for large-scale production, it is essential to identify or develop new cost-effective biocompatible and sterilizable materials [80] that are able to assure accuracy, precision and high quality during device development and after post-processing/finishing procedures [81,82,83]. However, research and development studies in the medical sector [84,85] must be addressed to meet all of the governance requirements [86].

Then, starting from the design of the final application, it is possible to work with many different materials [87,88], depending on the fabrication techniques. Hence, 3DP enables significant improvements in many fields, among them biomedical research, with a growing number of researchers in biomedical engineering that exploit 3DP in clinical applications [87,89].

In complex surgeries, 3DP can provide customized models on which the surgery team can train and improve pre-operatory planning. Three-dimensional printing also has the potential to train new surgeons by overcoming the observational learning method in favor of trial and error [90], with a reduction in wound complications, shorter recovery times and less post-operative pain [91,92].

Moreover, 3DP is a valid approach in the production of bioscaffolds for regenerative medicine, meeting the growing demand of human tissues and organs, upregulating regenerative cellular behaviors and improving cell attachment, proliferation and differentiation [93,94].

However, one of the main field of application of 3DP is represented by orthoses, that show clinical effectiveness through biomechanical and functional parameters and interesting results in terms of pain reduction, comfort, quality of life and patient satisfaction, and they are useful both in hospitals and in domestic rehabilitation programs [95,96,97,98].

In fact, the advantage of 3DP orthoses lies in the possibility of customizing orthoses, providing an individualized shape that can adapt to complex anatomical curves with optimal pressure points, a high ventilation surface, good aesthetics and a light weight [98]. Finally, by combining the EMG surface signals with the technology of 3DP exoskeletons, it is possible to set up a new personalized device to support the recovery of impaired movement functionality of chronic stroke patients, especially with regard to the upper limbs.

Moreover, the augmented design possibilities can meet the needs of subjective treatment, also in pediatric or complex cases [99], and improve treatment compliance (comfort and satisfaction [100]) via a patient-centered process design. In terms of cost-effectiveness, additive manufacturing minimizes waste by loading the printer with the precise quantity of material required to build the device [101,102], except for the supporting structure. Also, complex designs do not impact the production cost, as it is mostly dependent on the volume of the parts [16]. Moreover, additive manufacturing is not only beneficial to the environment, but it also leads to significant cost reductions due to the raw material savings. Furthermore, polymers, like polypropylene, PLA and ABS, which match the high stiffness requirements for orthoses can be recycled into new filaments [103].

Energy consumption in 3DP processes is influenced by several factors; for example, an increasing resolution implies extending the production time, resulting in higher energy consumption [103]. In this context, optimizing the design for additive manufacturing could lower energy utilization, production time, waste and human involvement [16]. However, other studies have reported that additive manufacturing results in lower energy consumption in comparison with traditional machining processes [104,105].

On the other hand, 3DP orthoses present some weaknesses: although the time of production can be shorter in comparison to traditional devices [99], the cost of the 3D printer, the software, the maintenance and training users could make it less affordable. In addition, the model design and FEM simulation play a pivotal role in preventing warping and brittleness, typical problems seen with many of the currently available bio-plastics [98], which are useful for their environmental sustainability (i.e., PLA printed without a heated chamber). A detailed description of the 3D printing orthosis production workflow is given in Table 1 [69].

In conclusion, 3DP orthoses have the potential to replace traditional technology thanks to cost-effective [15] personalization, requiring agreement on the role of 3DP in medical device fabrication [106] such as upper limb orthoses. Therefore, with the present systematic review, we sought to investigate the improvement generated from 3DP in upper limbs orthoses manufacturing in terms of upper extremity function, compliance, grip performance, pain, spasticity, oedema, ADL and satisfaction of stroke patients. We will analyze the technical features of the material utilized in orthosis manufacturing and the cost-effectiveness of the 3DP process in stroke patients.

## 2. Materials and Methods

### 2.1. Search Strategy

This systematic review was conducted and reported according to the Preferred Reporting Items for Systematic Reviews and Meta-analysis for Network Meta-analysis (PRISMA-NMA) guidelines and the Cochrane Handbook for Systematic Reviews of Interventions [107] and involves three electronic databases: PubMed, Web of Science and Scopus (from inception to 14 May 2023), following the search strategy of Table 2.

The aim of this review is to investigate the potential of 3D printing technology in the rehabilitation of stroke survivors, focusing on the improvement of upper limb function. PROSPERO registration: CRD42017065491.

### 2.2. Selection of Articles

We utilized the PICO (patient/population, intervention, comparison, outcome) approach: (P) post-acute, subacute or chronic stroke patients; (I) 3D-printed orthosis; (C), comparison between rehabilitation with 3D-printed orthosis and conventional rehabilitation, comparison between rehabilitation with 3D-printed orthosis and conventional orthosis (non-3D-printed orthosis); (O) preventing spasticity complications, improvement of the quality of life, pain and edema. Additional relevant trials were identified by manually reviewing the reference lists of the selected publications.

We included trials that met the following inclusion criteria: stroke survivors (acute, sub-acute or chronic stroke); 3DP orthosis; upper limb rehabilitation; RCT (randomized controlled trial); case control study; feasibility study as study design; full-text availability; written in English. The exclusion criteria comprised studies concerning lower limb rehabilitation in which only healthy subjects were included and which focused on cognitive conditions (e.g., mild cognitive impairment, dementia, psychiatric disorders) or those which did not focus on stroke (e.g., Parkinson’s disease, spinal cord injury, multiple sclerosis, traumatic brain injury, pain, cerebral palsy).

### 2.3. Data Extraction

The articles were independently screened by title and abstract by 3 reviewers. The full texts of the articles that were unclear from the title or abstract were reviewed in accordance with the selection criteria. Three reviewers extracted the data from the selected studies on a Microsoft Excel sheet. In case of disagreement, consensus was achieved with a fourth reviewer. Data extracted from each paper included the author and year of publication, subject characteristics (age, sample size, time post-stroke), descriptions of intervention (3D-printed orthosis) and outcome measures.

### 2.4. Quality Assessment

We used a modified variant of the STROBE criteria to conduct the methodological evaluation, using the ten criteria detailed in 3.10. Two authors independently assessed the score, and disagreements were evaluated and resolved by consensus. A numerical scoring system (1 if present; 0 if not present) was used to rate the items, and the studies were classified as follow: high risk of bias, score < 6; low risk of bias, score > 6. Then, to carry out the clinical review, we examined previous guidelines, research queries, adequate evidence, study quality, synthesis of results and their correct interpretation [108].

## 3. Results

### 3.1. Evidence Synthesis

The literature search generated 218 articles: 32 items from Pubmed, 78 from Scopus and 108 from Web of Science. Eighty documents were excluded for being duplicates and 138 articles were screened. Finally, 16 articles fulfilled the inclusion criteria; we excluded 6 articles after the quality assessment [108]. Lastly, 10 studies were included in the systematic review (Figure 1).

### 3.2. Synthesis of the Results

A total of 109 patients (ranging from 25 to 77 years of age; 83 men and 26 women) were included in the research analyses. A summary of the selected articles is shown in Table 3.

Three trials were RCTs [95,109,110]. Nine trials included chronic stroke patients (onset ≥6 months) [4,11,95,109,111,112,113,114,115], and one included both post-acute (within 3 months) and chronic stroke patients [110]. Two trials included both ischemic and hemorrhagic stroke [110,111], and nine trials did not report the etiology of the stroke. The side of the stroke (left/right) was reported in all selected articles except one [113]. One trial recruited patients from a hospital [110], one trial enrolled patients from a rehabilitation clinic [109,116], and eight trials did not report where patients were recruited.

### 3.3. Intervention Protocol

The rehabilitation methods varied greatly in terms of the length, frequency and volume of training.

Huang et al. tested the effectiveness of the 3DP orthosis in a task-oriented approach. Patients were divided into two groups (interventional group and control group), both completing a 4-week protocol and 2-week follow-up consisting of 30 min of therapist training and 30 min of daily home training. Both groups received the same amount of training, and the intervention group wore the orthosis throughout the protocol, while the control group was trained without the device [95].

Zheng et al. tested the 3DP orthosis and typical thermoplastic devices supported by the intervention and control groups, respectively. Both groups received the selected orthoses, for 4 to 8 h per day for 6 weeks, in order to perform the conventional rehabilitation therapy [110].

Yang et al. tested the 3DP orthosis in a six-week regimen. Both groups underwent conventional rehabilitation for 30 min, three times per week. The experimental group wore the 3DP brace for 6 h a day at home, while the control group had no brace and was advised to perform a series of independent handgrip exercises at home [109].

The remaining eight studies were all feasibility studies.

Chen, Zhou and Ben Abdallah tested the device efficacy after a rehabilitation training period (of variable duration) characterized by a series of exercises carried out independently at home by patients [11,111,113].

The other authors [4,112,114,115,116] tested the effectiveness of the devices before and after wearing.

### 3.4. Side Effects

Studies regarding robotic or exoskeleton hand orthoses set a range of motion limit in order to prevent excessive hyperextension stress in patients with anatomical limitations, such as spasticity and range of motion deficits due to muscle fibrosis or myotendinous retractions. In these cases, applying excessive hyperextension force can lead to injuries that can even worsen the state of spasticity [113,114].

Three studies reported no safety issues regarding increased spasticity and skin damage [110,112,115].

Two studies [4,110] assessed safety as an item on the QUEST scale; one study reported that the device received a high satisfaction score for safety [4].

The other trials did not evaluate side effects.

### 3.5. Technical Features

The technologies principally used in the selected studies belong to the material extrusion family (ASTM 52900-15), which is recognized as being a more flexible, not expensive, ergonomic (reduced weight) and sustainable (reduced waste) fabrication approach. Accordingly, eight of the ten described studies exploited the related benefits, and seven of them used PLA or TPU materials to build dedicated electronic cases too. The remaining experiments adopted selective laser sinter technology to fuse TPU and PA medical grade powders without adding toxic fillers, while Chen et al. described medical devices obtained from photo/thermal active polymers via CLIP (continuous liquid interface production) and MJP (multi-jet printing) technologies, respectively.

In detail, three authors developed via 3DP only the splint or the finger mechanisms [109,111,113]; three studies described all the printed pieces assembled on the orthoses trainer [11,114,115]; and the last four papers implemented the 3D hand shape in the final support design (Table 4) [4,95,110,112].

Ben [43] and Yang [109], via FDM (fused deposition modeling) technology, printed the final devices by using polylactic-acid (PLA) and acrylonitrile-butadiene-styrene (ABS) materials, respectively, while, Zhou [111] developed (robocasting) a silicone 3D-printed elastomeric finger to build a composite splint coupled with polypropylene plastic rings.

Chen [11] reached a weight of 200 g via PLA material and Park [114] reported a flexible device having 454 g of weight via TPU (thermoplastic polyurethane) material, both via FDM technology.

Finally, to assure patient safety and ergonomics with materials directly in contact with the skin, Dudley [4] reported the use of PLACTIVE material, a commercially available filament made from PLA enriched with antibacterial copper nanocomposite filler.

Huang [95] developed (FDM) an orthosis through 3D scanning, and Toth [112] developed a robotic orthosis coupling a thermoplastic polyurethane (TPU) thermal insulator with smart memory alloys as the active parts.

Laser-based technologies were also used by Toth [112] to fabricate the passive framework (weight without battery is 120–160 g, biocompatible polyamide (PA 2200), printed by SLS), and Zheng [110] reported the use of light-activated resin for the 3DP orthosis. Huber [115], to develop a dynamic hand brace, evaluated five different printable elastic actuators from TPU and elastomeric polyurethane EPU printed by different techniques (selective laser sintering (SLS), continuous liquid interphase production (CLIP), multi-jet printing (MJP)).

Three-dimensional-printed devices achieve the same elastic modulus of not-printed ones via SLS; MJP-processed actuators reach values between 87.12–94.1%, while CLIP technology does not overcome 7.27% compared with the standard TPU finger-extension component.

### 3.6. Outcome Measures

Upper limb function has been assessed using validated scales such as the manual function test (MFT) [112]; Fugl-Meyer Assessment for upper extremity scale (FMA-UE) [4,95,109,110,111]; box and block test (BBT) [4,95,111,115]; and action research arm test (ARAT) [11,111]. Moreover, the authors tested range of motion [110,113,114,116]; grasp performance with different sized objects [113,114]; hand force (grasp force, lateral pinch force, palmar pinch force) with dynamometer [11,95,111,115] and sEMG sensors [113]; and flexor and extensor strength during maximal voluntary contractions using a dynamometer [4].

The rate of perceived exertion (RPE) scale was used to evaluate exertion during the BBT test [4]. The modified Ashworth scale (MAS) was utilized for spasticity evaluation [109,110,111], the visual analog scale (VAS) for pain [110] and a four item swelling score for edema [110].

ADL were examined by some authors through the use of a range of tests such as pencil holding, eraser holding, handle grip, vertical handle, opening door handle, tool holding, plastic cup holding [112] and the Cedoke Arm and Hand Activity Inventory (CAHAI-9), a scale consisting of nine ADL items [114].

Satisfaction was assessed using the Likert scale [112,115]; OPUS questionnaire [114]; Quebec User Evaluation of Satisfaction with Assistive Technology (QUEST) [4,110]; and system usability scale [4]. User experience was evaluated in [109] through a subjective questionnaire regarding pain, spasticity, ease of self-wear and satisfaction.

Wearing time during rehabilitation programs was evaluated in [11,95].

### 3.7. Upper Limb Function

Toth et al. [112] reported a significant improvement in MFT scores before and after orthosis use, with an average improvement of 8.83 points (*p* = 0.00012). Differences were also found in handle grip (all participants could not perform the task without the orthosis and could after wearing it; *p* = 0.0011), tool holding (*p* = 0.0066), vertical handle gripping (*p* = 0.0011), plastic cup holding (*p* = 0.0303) and opening a door handle (*p* = 0.0303). Eraser holding was the only activity that did not improve (*p* = 0.1212).

Park et al. [114] reported an increase in the ROM of the MCP and PIP joints of the index finger (9.0° ± 12.1° and 22.8° ± 18.2°, respectively). The orthosis improved grip strength (max grip strength 26.27 ± 30.90 N), allowing patients to grasp common objects. Subjects were unable to complete the tasks in CAHAI-9 without the orthosis and were able to complete six out of nine tasks while wearing the device, with an average score increase of 14.40 ± 1.78.

Huang et al. [95] used FMA-UE as a cut-off point to categorize patients in the two groups. The number of blocks moved in the BBT increased significantly (100% increase in experimental group, 25% in control group; *p* = 0.034) after the 4-week training program, but only the patients in the experimental group maintained their improvement during the 2-week follow-up period (*p* = 0.042). Palmar pinch force improved significantly during the program and follow-up (204% in exp. group; *p* = 0.042 and *p* = 0.041) in contrast with no significant change in the control group. Lateral pinch force increased significantly (85.2% in exp. group and 13.6% in control group; *p* = 0.039) in both the experimental and control groups. Grasp force improved in both groups, showing better results in the 3DP (29.1% against 6.6%) orthosis group.

Dudley et al. [4] evaluated the FMA with a score of 0 without the orthosis and 10 while wearing the exoskeleton. The BBT resulted in five more blocks being moved with the aid of the device. The RPE decreased while using the exoskeleton (4.5 before and 8 with the device). EMG extensor activation was greater with the device (39.09% and 20.24% without the device).

Ben et al. [113] showed that the 3DP device had a positive effect on ROM and hand ability to perform simple tasks.

Chen et al. [11] found a significant improvement in grip force (63.3 N mean value pre-test, 86.1 N mean value post-test, *p* = 0.028) and lateral pinch force (24.6 N mean value pre-test, 28.9 N mean value post-test, *p* = 0.028); palmar pinch force improved but did not reach statistical significance (*p* = 0.068). The participants’ ARAT score increased from 20.33 to 31.33 (*p* = 0.026).

Huber et al. [115] found that pinch force decreased with higher elastic modules, as did pinch aperture force. Moreover, a significant decrease in the BBT score was found with high elastic modules. Interestingly, the commercial brace reduced hand function more than the 3DP device.

Yang et al. [109] assessed the participants four times: Pre1 before wearing the device, Pre0 immediately after wearing it for the first time, Pos3 after three weeks of intervention and Pos6 after six weeks. In the experimental group, there was a significant decrease in the MAS score for the finger flexor at Pos3 (Pre0 3.2 (0.6), Pos3 2.4 (0.9), *p* = 0.03) and at Pos6 (2.0 (0.6) *p* < 0.01 for wrist flexors) when compared to Pre0. No significant changes in MAS were found in the control group; a slow decreasing trend was observed. The FMA-UE score improved significantly at Pos3 (Pre0 27.8 (11.7), Pos3 30.5 (12.0), *p* = 0.02) and Pos6 (36.8 (11.2) *p* < 0.01) in the experimental group. In the control group, a slight improvement was found at Pos3 and a significant increase at Pos6 (Pre0 21.6 (11.5), Pos3 23.0 (12.7) Pos6 26.0 (14.7), *p* = 0.01).

Zheng et al. [110] observed that after six weeks, 65% of patients in the intervention group showed spasticity relief as opposed as 30% in the control group (*p* = 0.02). Changes in extension (mean increase of 10.3° (3.8°) vs. control group 3.3° (2.9°); *p* < 0.001) and ulnar deviation angles (mean increase 5.3° (6.0°) vs. control 1.1° (2.6°); *p* = 0.028) were significantly higher in the experimental group, as with the FMA-UE (1.3 (1.0) score improvement vs. control 0.2 (0.3)) and swelling scores. No statistically significant between-group differences were found for flexion and radial deviation angles, changes in VAS scale and subjective sensation score.

In Zhou et al. [111], after 20 rehabilitation sessions, FMA improved from 26.75 ± 8.73 to 33.50 ± 8.70, BBT from 1.75 ± 2.36 to 5.50 ± 5.80 and the ARAT score from 20.00 ± 14.35 to 29.45 ± 15.71. The MAS score improved in two subjects out of four.

### 3.8. Satisfaction and Motivation

Patients were satisfied with the use of the orthosis in [112,114] (OPUS score 4.3 ± 0.48).

Moreover, the device was considered sufficiently lightweight (OPUS score 4.3 ± 0.07) and comfortable (OPUS 4.2 ± 0.5) in [114].

In Yang et al. [109], participants in the intervention group showed a significant difference compared to the control group in terms of satisfaction (*p* < 0.01), ease of use (*p* < 0.01) and wearing time (*p* < 0.01). Pain ratings showed no difference between the two groups, with participants still experiencing mild annoying pain.

In Huang et al., the authors analyzed the time of use in both groups, showing more training time in the 3D-printed device group (56.8 min experimental group and 26.4 min control group) [95]; in Chen et al., participants spent an average time of 63 min, 23 min more than the minimum recommended training in rehabilitation settings [11].

Average scores of 4 on QUEST (“quite satisfied”) and 90 out of 100 on the SUS scale were obtained in [4].

Huber et al. compared a 3DP orthosis with a commercial orthosis and found no significant difference in the usability categories between the two devices [115].

### 3.9. Limitations

Many studies reported a small patient sample as a limit to their research [11,95,110,113,115]. In addition, many of the trials featured a rehabilitation intervention that was shorter than the conventional long-term rehabilitation program that stroke survivors usually go through, making the benefits less noticeable [11,115].

In Park et al., the shape of the orthosis proved to be a limitation in completing two tasks (fastening five buttons, cutting medium resistance putty) in the CAHAI-9 protocol chosen to evaluate its performance. The authors also report that the tension cables can break during use [114]. In Huang et al., the authors found that occupational therapist assistance was required to adjust the cable tension, making it difficult for the orthosis to be worn independently by the patient [95]. In Dudley et al., the authors developed a device that completely encloses the fingers. The lack of proprioceptive feedback may have affected the study, most notably when subjects performed the BBT [4]. In Ben et al., the authors found that the device was not suitable for independent work by patients at home, and that a therapist was required to evaluate and provide motivation for the training protocol. In addition, they found the orthosis to be more suitable for advanced stroke survivors and recommended conventional rehabilitation therapy for more recent strokes before starting to use their device [113]. In Chen, the authors reported a possible resistance problem due to the material chosen to print the orthosis; for this reason, they only recruited patients with a MAS tone level of less than two. After the trial, no breakages were reported, suggesting that the device may be suitable for sustaining a higher tone [11]. In some cases, material surface texture and thermal conduction can deliver unique sensory afferent information to the user, a scenario that could affect motor control, leading to the facilitation or inhibition of the user’s pinch force [115]. Yang et al. reported that the results were limited to stroke survivors with mild to moderate upper limb disability. The dynamic splint can only be worn by stroke survivors who can extend their wrist and open the affected hand with passive movements [109]. Finally, the issue of using subjective questionnaires or not entirely objective scales or measures to assess outcomes has been raised by some authors [109,110].

### 3.10. Study Quality

Four of the ten studies considered (40%) [11,95,109,114] were of excellent quality, three were of very good quality (30%) [4,112,115] and the remaining (30%) [110,111,113] were of good quality, as shown in Table 5.

## 4. Discussion

Stroke is the one of the leading causes of death and long-term disability worldwide. Patients affected by hemispheric stroke complain of a reduction in their upper limb motion ability in approximately 80% of the cases in the first days after stroke; that persists in 30% to 66% of cases at 6 months and is associated with higher levels of anxiety and reduced perception of health-related life quality. The improvement of upper limb function plays a role in stroke rehabilitation [117]. In this context, rehabilitation has been shown to be beneficial for stroke patients, and the correct personalization of orthoses for daily life activity avoids conventional orthoses cutaneous and circulatory complications caused by muscle atrophy and joint stiffness and the consequent discomfort that limits their use.

This review investigated the role of 3D-printed upper limb orthoses in the rehabilitation of stroke patients, focusing on the technological advances and solutions of personalized orthoses and active exoskeletons.

The studies reported a greater improvement in patients treated with 3DP upper limb and hand orthoses in terms of grip force and motor functions [20,95,110,118].

Park et al. [114] reported an improvement in patients wearing the personalized orthosis while performing a series of ADL tasks. These results are in accordance with the results of Chen et al., showing increases in strength and hand functionality after use for 4 weeks with a 3DP orthosis [11]. By contrast, Huber et al. [115] determined that hand function is linked to the material utilized for 3DP; in particular, the increase in the elastic properties of the materials is associated with a decrease in grip force and hand opening.

Our review confirms the results of a recent article by Schwartz et al. examining the clinical use of 3D-printed orthoses for upper extremity musculoskeletal conditions such as fractures, wrist pain, overuse conditions or trauma. The authors evaluated outcomes such as range of motion, side effects, measures of grip and pinch strength, pain, measures of patient satisfaction (standardized and non-standardized) and patient functionality.

This improvement is achieved even in the neurological affection of the upper limb. In particular, all studies we included that used the 3D-printed orthosis showed a reduction in pain (assessed using the VAS scale), an increase in patient satisfaction in terms of comfort, ease of use, weight of the orthosis, time and independence in activities of daily living (assessed using the QUEST, OPUS and Barthel scales) and improved range of motion [18].

Furthermore, in patients with upper limb spasticity, the use of 3DP orthoses appears to have a positive effect on spasticity compared to the control groups [20,110,118], with interesting results in motor and sensory function in stroke patients and gross manual dexterity. In accordance with a recent scoping review by Oud et al. revealing a statistically relevant improvement in spasticity and movement pattern after using a 3DP orthoses, 3DP orthoses combined with a rehabilitation program can give a better outcome on ROM, motor function and spasticity than thermoplastic orthoses [110,118].

The ability to customize orthoses is one of the main advantages of 3DP. In fact, it is possible to create a personalized shape for each patient that can adapt to complex anatomical curves with optimal pressure points, a high ventilation surface and good aesthetics [98,119]. However, no significant differences were found in terms of pain when evaluating the VAS scale in RCTs and comparing 3DP orthoses and conventional devices [95,109,110].

Regarding manufacturing, the traditional method of standard orthoses personalization is relatively long. It is necessary to manually correct the shape and dimensions of the orthosis depending on the patient’s anatomy [15,120,121]. Furthermore, it is difficult to produce multiple customized orthotics of the same quality and complex design. Moreover, while a manually produced orthosis takes approximately 1 week, without considering the time required for the 3D design, the time to produce a 3D-printed orthosis varies from 12 to 23 h [95,122,123,124]. On the other hand, one of the most controversial arguments about the application of 3DP in clinical practice is represented by the high cost of 3D printers, software, maintenance and training of healthcare professionals that could require a high initial investment, but the long-term production could make the implementation of this technology cost-effective. In particular, when considering more complex designs such as exoskeletons or robotic orthoses, 3D printing could be a valuable solution [4,115,125].

According to Keller et al. [19], a well-organized workflow and the interconnection between the different departments (e.g., emergencies, engineering, orthopedics and rehabilitation) can make 3D printing orthoses a feasible solution in hospital settings. For example, 3D data acquisition and design could be sped up, outsourcing to medical engineers or automating the procedure through dedicated software. However, it is not always easy to create 3D printing models. The process of capturing images requires an initial scanning of the affected segment by using a 3D scanner. Nevertheless, in patients with musculoskeletal retractions or dyskinesia, incorrect posture or continuous movement does not allow the setting of the correct scanning procedure, representing an important limitation. In this case, it is possible to use a posture corrector or have multiple scans conducted by an expert technician [120,126,127]. The same 3D file can be reused and reworked later without the need to re-capture the patient’s anatomy [128].

One of the main limitations of 3DP is the manufacturing of large sizes of orthoses determined by the limited capacity of most common printers [15,19]. However, this barrier can be overcome by printing modular devices or by joining segments together.

On the other hand, the results show that the comfort and satisfaction of patients wearing 3DP orthoses is higher than patients who have used traditional orthoses [109,112,114,116]. This could be related to the light weight of the materials used, the presence of ventilation holes, and the reduced swelling of the limb due to the better fit [110,112]. This appears to be associated with increased patient compliance, confirmed by three authors reporting that patients spent more time exercising when 3DP devices were involved in their rehabilitation program [11,95,109]. The high feeling of the 3DP orthoses is in line with the results of Shiwei et al., who investigated the differences between traditional plaster-molded foot orthoses and 3DP orthoses in a population of female asymptomatic runners with excessive foot pronation [129], which resulted in better perceived comfort in comparison with the control group [127].

The most commonly used materials in 3DP are plastic polymers; therefore, lighter devices can have problems with brittleness and warping. This is more evident in lower limb orthoses where devices must withstand continuous mechanical stress during walking [98]. A commonly used material for 3DP orthotics is polylactic acid (PLA) printed through the FDM technique. PLA is a material that contains no heavy metals or pollutants. For this reason, PLA is an environmentally friendly, renewable and biocompatible material. Its disadvantages are inferior mechanical strength and stiffness properties [130], such as brittleness, poor heat stability, low crystallization, low elongation at break, poor impact resistance and a low heat deformation point. However, various additives, e.g., cellulose, glass fiber, carbon fiber, metal-based compounds or carbon nanotubes, can be incorporated into PLA to improve its properties, forming nanocomposites with better mechanical and thermal features and also additional features such as electrical conduction and antibacterial action [131].

A further cause of fragility in 3D-printed parts is the anisotropic print direction in the fused deposition modeling process. This limitation can be solved by choosing a layer stacking scheme or changing the printing direction to achieve a better load distribution. In addition, alternative 3D printing techniques such as selective laser sintering, material jetting and vat photopolymerization can be integrated to achieve isotropic strength in printed modules [16].

In this context, to reduce the environmental impact of 3D printing orthotics, the disposal and recycling of devices and scrap material must be considered. After cleaning and shredding, polymers can be re-extruded into filament. However, this process is still far from being cost-effective, and successive heating cycles and residual impurities reduce the physical properties of the filament, which may not be desirable for the production of an orthosis [131,132]. However, recycled PLA can be mixed with virgin PLA to achieve good mechanical features close to the original material [132]. Interestingly, carbon-impregnated PLA mixed with virgin PLA shows increased mechanical properties [133]. In a recent review, Voet et al. collected all the recyclable powder polymers developed for SLS printing, and Guoqiang Zhu et al. developed recyclable and reprintable castor oil (CO)-based photopolymers for DLP [64].

Moreover, other polymers, e.g., ABS, polycarbonate, polypropylene, high- and low-density polyethylene, polystyrene and polyethylene terephthalate, are commonly recycled [132].

The growing scientific interest in the production of 3D-printed orthoses is relatively recent, and the included studies have all been published in the last six years.

A strength of this review is represented by the specific condition of the included patients, which gives more specific insight into the impact of 3D-printed orthoses on stroke survivors.

The results show how 3DP devices improve upper limb functionality and strength, with important implications for ADL improvement and user satisfaction in stroke patients. In addition, the ability to customize the orthosis allows for greater comfort and prolonged use, exceeding the limitations of traditional orthoses. These results pave the way to the integration of 3D printing in rehabilitation.

However, the current literature on this topic is represented by a small number of studies conducted on a small number of patients, and future research is needed to improve the accuracy and efficiency of additive manufacturing for capillary diffusion in the medical field. In this context, artificial intelligence could play a role in the customization process to improve orthosis design based on patients’ needs. Moreover, the development of new biocompatible materials, absorbable materials and printing techniques could allow a wider utilization in a clinical setting with respect to environmental pollution [69].

## 5. Conclusions

Three-dimensional printing technologies enable the production of complex medical devices to meet customization costs and patient needs, reducing risks, improving patient outcomes and reducing lead times via delocalized productions.

Moreover, considering the negative effect of stroke on upper limbs, 3DP orthoses have shown interesting results in terms of functional improvement, satisfaction and cost. Three-dimensional printing could, therefore, represent a cost-effective and valid solution in a clinical setting, with the possibility of designing upper limb orthoses tailored to the patient’s needs which consequently have to be progressively introduced inside medical SOPs (standard operating procedures) during rehabilitation.

Unfortunately, standardization, regulatory compliance, quality control and material limitations, such as the utilization of AM families for sensors or wearable devices, are challenging to address.

The materials used in AM must be biocompatible, sterilizable and safe for human use, assuring strength, flexibility and wear resistance for long periods (without losing performance levels). Moreover, medical device production costs (disposables, materials, tools, software), and the related accuracy, are still associated with the technologies used and the post-processing, consequently reducing the fast scalability of skilled labor able to produce medical devices.

## Figures and Tables

**Figure 1 bioengineering-10-01256-f001:**
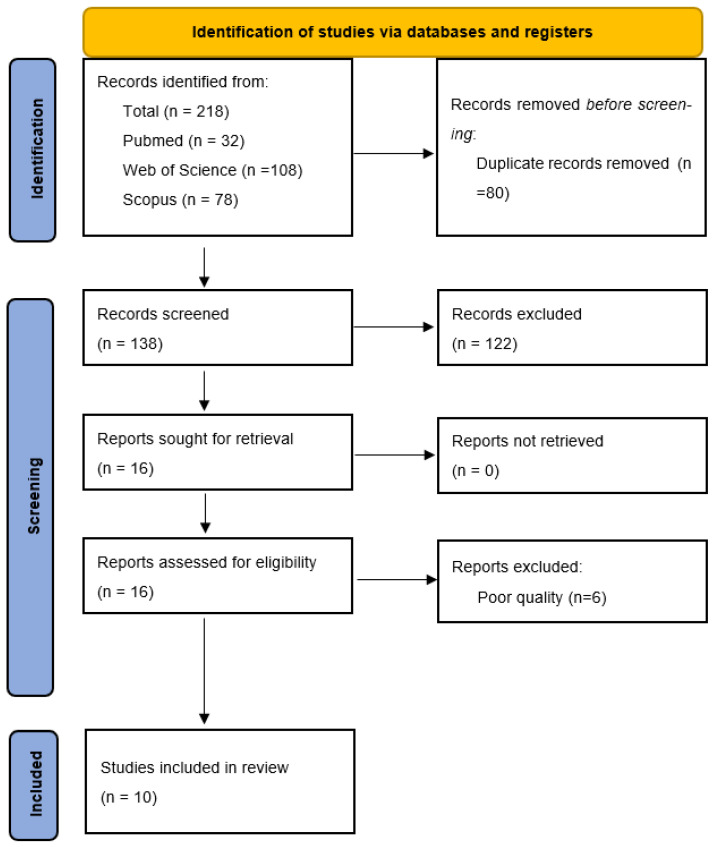
PRISMA flowchart.

**Table 1 bioengineering-10-01256-t001:** Workflow for upper limb orthosis production.

3DP Orthoses of the Upper Limb	Traditional Personalized Upper Limbs Orthoses
Production takes an average of 24 h.	Production takes more than 2 days.
Measurement is automated and performed with a 3D scanner.	The measurement is performed manually.
Production has fewer steps: 3D body shape acquisition, CAD-CAE model construction, orthosis printing.	Production is more complex. It requires the acquisition of the body shape, the creation of the negative mold, manual fabrication, polishing and adjustments.
Tools needed: 3D printer; 3D scanner.	Tools needed: plaster; thermoplastic material; tools and machines for measuring and shaping.
Low rate of ergonomic failure	Modifications are often necessary when testing the product on the patient.
It is easy to reprint the orthosis with anatomical adjustments.	Post-production modification requires great skill; is not always possible and affects the quality of the product.

**Table 2 bioengineering-10-01256-t002:** Search strategy.

PubMed ((stroke) OR (cerebrovascular accident) OR (CVA)) AND ((3dp) OR (3d printing) OR (additive manufacturing)) AND (rehabilitation) Scopus TITLE-ABS-KEY (“stroke” OR “cerebrovascular accident” OR “CVA”) AND (“3dp” OR “3d printing” OR “additive manufacturing”) AND “rehabilitation” Web of Science (“stroke” OR “cerebrovascular accident” OR “CVA”) AND (“3dp” OR “3d printing” OR “additive manufacturing”) AND “rehabilitation”

**Table 3 bioengineering-10-01256-t003:** Main characteristics of the randomized controlled trials included in the present systematic review.

Author	Patient	Tools	Training	Intervention	Control Group	Assessment	Outcome
Yang et al., 2021 [109]	N = 25, 21 M/4 F Age: 45.7 ± 0 years. Stroke resulting in upper limb spastic hemiplegia more than 1 year before admission to the study.	Dynamic 3D-printed hand–wrist splint.	40 min, three times a week for 6 weeks.	Wear a custom-made, dynamic 3D-printed hand–wrist splint for at least 6 h per day at home for the 6-week intervention in addition to conventional rehabilitation therapy.	Did not wear a hand splint and were involved in a home exercise program in addition to conventional rehabilitation therapy.	MAS; FMA; questionnaire measured with a VAS regarding pain, spasticity, satisfaction, ease of self-wear, pain.	The 3D-printed dynamic hand–wrist splint was effective in reducing wrist and finger flexor spasticity. Also, there was a significant alleviation in self-reported spasticity after 6 weeks of intervention.
Zheng et al., 2020 [110]	N = 40, 31 M/9 F Aged 35–80 years. Ischemic or hemorrhagic stroke. Limb hemiplegia within 2–12 months of acute event.	3D-printed orthosis versus conventional thermoplastic orthosis.	Wear the orthosis at home for about 4–8 h per day for six weeks.	Conventional rehabilitation therapy with 3D-printed orthosis.	Conventional rehabilitation therapy with low-temperature thermoplastic plate orthosis.	MAS, PROM wrist, FMA, swelling scores, VAS, subjective feeling scores.	3D-printed orthosis showed greater improvement compared with low-temperature thermoplastic plate orthosis in spasticity and swelling, motor function of the wrist and passive range of wrist extension.
Zhou et al., 2022 [111]	N = 4, 2 M/2 F Stroke survivors with hand impairment.	New personalized 3D-printed soft robotic hand (SECA).	20 sessions (three times a week) of hand function rehabilitation training consisting of 45 min sessions with 5 min breaks to prevent fatigue.	Subjects were stimulated to practice hand closing and opening exercises. Moreover, they were required to conduct ADLs using the 3D-printed soft robotic hand.	No control group.	FMA, BBT, grip force, ADL.	Significant improvement in hand function in all subjects evaluated. Three subjects improved significantly in the BBT performance. Two subjects (50%) showed an improvement in spasticity.
Dudley et al., 2021 [4]	N = 1, 1 M Age 67 y.o. Stroke survivor with hand impairment.	A 3D-printed upper limb exoskeleton.	No training.	Hand function was assessed with and without the orthosis. Strength was tested during MVC using a dynamometer. BBT was performed with and without the device; exertion was measured by EMG signal and RPE scale.	No control group.	FMA, BBT, Borg RPE scale, SUS, QUEST.	While wearing the exoskeleton, the subject improved in flexion and extension and successfully performed three out of the four different grasps; the subject’s normalized EMG extensor activation was larger while using the exoskeleton compared to the control group. The reported RPE scores were lower when using the exoskeleton, reducing fatigue. The device received high scores from the QUEST and SUS surveys.
Huang et al., 2019 [95]	N = 10; 9 M/1 F Age: 59.6 ± 8.0 y.o. with hemiparesis of the upper limb. Acute event occurred more than 6 months previously.	3D-printed dynamic hand device (3D-DHD).	30 min of onsite training twice a week and at least 30 min of home training for the rest of the week for 4 weeks.	Traditional task-oriented approach training and 3D-DHD autonomous training at home.	Traditional task-oriented approach training.	BBT, FMA-UE, hand force (GF, LPF, PPF).	3D-DHD improved dexterity, pinch force and GF in individuals with chronic stroke. Because of the opposition-based design, the 3D-DHD group exhibited more improvement in terms of palmar pinch force than the control group. Participants wearing 3D-DHD showed higher motivation during training.
Toth et al., 2020 [112]	N = 6; 2 M/4 F Age: 42.0 ± 17 years Stroke survivor with hand impairment.	Personalized orthosis for post-stroke patients.	No training.	Performing functional manual tasks with or without the orthosis.	No control group.	Manual function test, daily living functionality tests and Likert scale.	The orthosis significantly increased the functionality of the users in all tasks evaluated, except eraser-holding.
Ben Abdallah et al., 2017 [113]	N = 2; 2 M Stroke survivor with hand impairment.	3D-printed hand exoskeleton.	Home exercise every day for 26 days with periodic therapist contact.	Not stated.	No control group.	ROM of hand joints assessment at baseline and post-intervention.	Positive effect on finger ROM and in the hand function. This technology also shows potential application in domestic rehabilitation.
Chen et al., 2022 [11]	N = 6; 6 M Age: 42.0 ± 15 years Acute event occurred more than 6 months.	3D-printed multi-functional hand device (3DP-MFHD).	Training at home for 4 weeks for at least 40 min per day, 5 days per week.	Training using 3DP-MFHD.	No control group.	Hand function evaluation through GF, LPF, PPF and ARAT.	The 3DP-MFHD significantly improved hand strength in terms of grip force and lateral pinch force in home rehabilitation.
Park et al., 2023 [114]	N = 10; 4 M/6 F Age: 64.5 ± 12.5 years Chronic stage stroke with limited active ROM.	3DP active assisting hand orthosis.	No training.	Performing tasks and movements with or without the orthosis.	No control group.	ROM, CAHAI-9, grasp performance.	The orthosis can successfully assist with grasping tasks in ADL by providing a sufficient grip aperture and grip strength. The orthosis increased the grip aperture and grip strength of all participants, enabling successful grasping even in severe degrees of spasticity.
Huber et al., 2023 [115]	N = 5; 5 M Age: 50.0 ± 0 years History of subacute or chronic stroke.	3DP dynamic hand bracing with 3DP elastic modules.	No training.	Assessing device usability and comparing with commercially available tools. Testing different tension elastic modules in different tasks.	No control group.	Stroke impairment scale, Likert-10 questionnaire, BBT, dynamometer.	Both devices tested did not show an improvement in hand function. Pinch force decreased more with higher elastic modules, as did pinch aperture force. A significant decrease in BBT score was found with higher elastic modules. The commercial brace reduced hand function more than the 3DP device.

ABBREVIATIONS: 3DP = 3D-printed/3D printing, MAS = modified Ashworth scale, VAS = visual analog scale, BBT = box and block test, PNF = proprioceptive neuromuscular facilitation, NDT = neurodevelopmental technique, GF = grasp force, LPF = lateral pinch force, PPF = Palmer pinch force, PROM = passive range of motion, RPE = rate of perceived exertion, MMSE = mini mental state exam, OT = occupational therapist, 3D-DHD = 3D-printed dynamic hand device, 3D-MFHD = 3D-printed multi-functional hand device, ARAT = action research arm test, MVC= maximal voluntary contraction, SUS = system usability scale, QUEST = Quebec user evaluation of satisfaction with assistive technology, y.o. = years old.

**Table 4 bioengineering-10-01256-t004:** Technology and material.

Technology	Material	Designed Part	Reference
FDM	PLA + nanoparticles	All the orthosis parts	Dudley et al., 2021 [4]
FDM	n/a (printing T = 190–205; hp: PLA)	All the orthosis parts	Huang et al., 2019 [95]
FDM-SLS	TPU flexible + PA	All the orthosis parts	Toth et al., 2020 [112]
n/a (SLA or DLP)	Resin (generic)	All the orthosis parts	Zheng et al., 2020 [110]
FDM	PLA	Splints and supports	Chen et al., 2019 [11]
FDM	TPU flexible + PLA	Splints and supports	Park et al., 2023 [114]
SLS—MJP—CLIP	TPU—TPU—EPU	Splints and supports	Huber et al., 2023 [115]
FDM	PLA	Splints	Ben et al., 2017 [113]
FDM	ABS	Splints	Yang et al., 2021 [109]
n/a (hp: robocasting)	Silicon	Splints	Zhou et al., 2022 [111]

**Table 5 bioengineering-10-01256-t005:** Quality Score.

Articles	Criteria for the Quality Scoring										Score
	1	2	3	4	5	6	7	8	9	10	
Ben Abdallah et al., 2017 [113]	1	1	0	1	1	1	1	1	0	0	7
Chen et al., 2022 [11]	1	1	1	1	1	1	1	1	1	1	10
Dudley et al., 2021 [4]	1	1	0	1	1	1	1	1	1	1	9
Huang et al., 2019 [95]	1	1	1	1	1	1	1	1	1	1	10
Huber et al., 2023 [115]	1	1	1	1	1	1	0	1	1	1	9
Park et al., 2023 [114]	1	1	1	1	1	1	1	1	1	1	10
Toth et al., 2020 [112]	1	1	0	1	1	1	1	0	1	1	8
Yang et al., 2021 [109]	1	1	1	1	1	1	1	1	1	1	10
Zheng et al., 2020 [110]	1	0	1	1	1	1	1	1	0	0	7
Zhou et al., 2022 [111]	0	1	0	1	1	1	1	0	0	1	6

Informative and balanced abstract. (1) Presence of detailed objectives, including the hypotheses of the study (2); available eligibility criteria (3); for variables of interest, availability of data sources and characteristics of measurement methods and description of correspondence of methods (if there are two or more groups) (4); quantitative variables (5); summarized characteristics of the study population (6); key findings focused on the study objective (7); description of limitations (8); careful interpretation of outcomes based on the objectives, related literature and further relevant evidence (9); funding statement (10).

## Data Availability

Not applicable.

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
