# Peer review of "The Upper Limb Orthosis in the Rehabilitation of Stroke Patients: The Role of 3D Printing"

_bioengineering, 2023, doi:10.3390/bioengineering10111256_

Round 1

Reviewer 1 Report

Comments and Suggestions for Authors

Generally, the manuscript has been written comprehensively and has no serious or critical issues to be mentioned. However, before the publication, it is better to consider some aspects more carefully which are mentioned below to enhance the manuscript and make it more sophisticated:

Comment1:

In the introduction section, it was mostly mentioned the background of the related fields and previous studies. It is great; but in the introduction especially the last paragraphs, it is suggested to be dedicated to mentioning the objectives of this article and the aim of the study.

Comment 2:

In sections 2 and 2.1, it is suggested to separate the search strategy and data synthesis and provide a separate subsection for data synthesis in order to distinguish these two scopes.

Comment3:

 The discussion section should cover three main divided aspects including a summary of findings, strengths and limitations of the study and the implications.

In the discussion section of this manuscript, the two mentioned parts of the summary of findings, and strengths and limitations of the study were covered comprehensively. However, it seems there is a lack of explanations for the last aspect, “implications” in comparison with the two others. it is suggested that authors explain more about the “implications” of this study and future potentials.

Comments on the Quality of English Language

 Minor editing of English language required

Reviewer 2 Report

Comments and Suggestions for Authors

The authors describe one of the main sequelae of stroke patients is the deterioration of motor functions, where orthosis plays a primary role in rehabilitating patients. This article focuses on the clinical study of upper extremity orthoses.  Likewise, the use of polymeric materials for the construction of these devices, using 3D printing, is mentioned. However, the study is not so deep from this point of view. For this reason I do not recommend this article for this journal.

1. The conclusion should be expanded regarding the use of materials in 3D printing, taking into account the advantages, disadvantages, and cost of these, as well as the impact of this technology on the development of these devices.

2. The article mentions the various clinical studies where 3D printed orthoses are applied; however, the mention of materials or 3D printing techniques is very poor. In contrast, the journal includes topics such as biomedical therapy, rehabilitation engineering, biomaterials, and biomedical devices. The subject of this article does not fall under those guidelines.

3. Lines 111 and 112 mention the use of bioplastics and the problems they may present to be used in 3D printing; however, they are not specific in what type of polymer and its difficulties.  

4. Line 206. What kind of anatomical limitations?

5. Lines 231-234 makes mention of 3D printing, however, the information is very ambiguous.  

6. Lines 251-254, the work team evaluated different printable TPU and elastomeric polyurethane materials under other synthesis techniques. However, no reference is made to the optimal results of the study. 

7. Line 389 does not define objective 1 in Table 4.

8. Lines 466-470 mention a prosthetic foot; however, only prosthetics for upper limbs were mentioned in the entire text. 

9. Lines 497-500 refer to the recycling of photopolymers, but they are only mentioned and do not indicate their use or exploitation.

10. The first reference used is an article produced by the author of this text.  In the same way, it is a member of the work teams of references 2 and 8; likewise, references 1, 2, and 34 belong to the corresponding author. On the other hand, only six items belong to materials. Finally, references 35 and 36, 5 and 43, 9 and 45, 51 and 53, are the same.

Reviewer 3 Report

Comments and Suggestions for Authors

The review covers an important problem of post-stroke rehabilitation. Specifically, it is devoted to the role of 3D printed devices for upper limb bio-motor function restoration.

As pointed out in the abstract, “3D printing provides the possibility produce customized, eco-sustainable and cost-effective orthoses”. These properties should be elaborated in the paper. In the introduction, the authors give some details about customization, but nothing or almost nothing is said about eco-sustainability and cost-effectiveness. Probably other manufacturing methods (that are less eco-friendly and cost-effective) should be mentioned for comparison.

In the abstract, the authors state that “The aim of this study was to highlight how 3DP could overcome the limitations of standard medical devices”. It should be specified what standard devices and what limitations are meant? In the 10 papers which were included to the meta-review, only one seems to use a device which is called “low-temperature thermoplastic plate orthosis.”. Other papers use “standard rehabilitation protocol” or no control group.

Overall, the paper is well written, methods and conclusions are clear.

Comments on the Quality of English Language

Overall, the paper is well written

Round 2

Reviewer 2 Report

Comments and Suggestions for Authors

The authors followed all the recommendations, so the manuscript is approved